# Assessing the Impact of Sepiolite-Based Bio-Pigment Infused with Indigo Extract on Appearance and Durability of Water-Based White Primer

**DOI:** 10.3390/ma17040941

**Published:** 2024-02-18

**Authors:** Massimo Calovi, Stefano Rossi

**Affiliations:** Department of Industrial Engineering, University of Trento, Via Sommarive 9, 38123 Trento, Italy; massimo.calovi@unitn.it

**Keywords:** sepiolite powder, indigo extract, bio-based pigment, waterborne paint, green coating

## Abstract

The objective of this study is to evaluate how two varying amounts of sepiolite-based powder, infused with indigo extract, affect the appearance and durability of a water-based, white primer. To examine the influence of this eco-friendly pigment on the coatings’ overall appearance, assessments were performed for color, gloss, and surface roughness. Additionally, the coatings were investigated through optical and electron microscopic observations, to evaluate the distribution of the pigment within the polymer matrix. The effect of the pigment on the coating’s durability was assessed through accelerated tests, including exposure in a salt spray chamber and a UV-B chamber. These tests aimed to evaluate the emergence of defects and changes in the appearance of the samples over time. Furthermore, the impact of different quantities of sepiolite-based powder on the coating’s ability to act as a barrier was assessed using liquid resistance tests and contact angle measurements. These evaluations aimed to understand how the coating responded to various liquids and its surface properties concerning repellency or absorption. In essence, this study underscores the considerable influence of the eco-friendly pigment, demonstrating its capacity to introduce unique color and texture variations in the paint. Moreover, the inclusion of the pigment has enhanced the coating’s color stability, its ability to act as a barrier, and its overall durability when exposed to harsh environments.

## 1. Introduction

The previous unsustainable linear economy caused raw material prices to surge and led to irreversible environmental harm, resource depletion, and a buildup of waste [1]. As a response, a contemporary regenerative economy is emerging, emphasizing a circular production and consumption system to curtail environmental effects [2]. This includes replacing petroleum-derived goods with renewable sources like bio-based materials, reflecting a focused effort on improving waste management systems. In this context, incorporating natural-origin additives and pigments into composite materials and coatings is gaining significant attention from both scientific and industrial perspectives [3,4]. Industries are constantly seeking ecological and versatile alternatives to conventional synthetic fillers and additives [5,6], which often lack considerations for environmental sustainability in their production processes [7]. Recently, fillers sourced from food and natural waste have demonstrated their ability to enhance the value of composite products, lowering manufacturing expenses and rejuvenating recycled materials [8]. For instance, the utilization of bio-based materials in manufacturing rose from 5% in 2004 to 12% in 2010 and climbed to approximately 18% by 2020, with estimates projecting a further 25% increase by 2030 [9].

These ideas have spurred both research and industrial sectors to consistently utilize diverse, bio-based fillers sourced from nature or agri-food waste in polymeric matrices [10]. For instance, chicken eggshells (ES) stand out as one of the extensively employed bio-based resources for filling organic coatings [11,12,13] due to their abundant availability as a bio-waste form and their high calcium carbonate content (95%) [14]. Likewise, lignin and cellulose serve as renewable resources used as strengthening nanofillers in composite materials [15,16]. Additionally, the diverse range of discarded seashells from mollusks offers an endless supply of CaCO_3_, their primary component, which can serve as an alternative to conventional calcium carbonate in eco-friendly coatings [17,18,19]. Moreover, the agro-industrial sector generates substantial waste that holds the potential for functional use as cost-effective, high-performance fillers. Within these resources, olive pit powders [20,21], almond shells [22], pistachio nutshells [23], apricot and argan remnants [24], cherry seeds [25], peanut shells [26], avocado seeds [27], and rice husks [28] have been employed to enhance the various properties of polymer matrices. Similarly, numerous studies have examined the impact of bio-based resources as environmentally friendly additives to enhance the hydrophobic characteristics of organic coatings. This exploration involves the utilization of natural waxes [29,30] and animal proteins [31], among other green alternatives.

Absolutely, aesthetics have become a crucial aspect of coatings alongside durability. As a result, numerous recent studies have focused on exploring natural and bio-based pigments. The aim is to align with principles of the circular economy and decrease the environmental footprint associated with new coloring additives. For example, spirulina [29,32] and turmeric [30] extracts have demonstrated impressive durability and promise when incorporated into wood paints. Numerous studies have focused on examining the dyeing potential of leftover fruit and vegetable materials [33], as well as assessing how *Aspergillus carbonarius* can be used in processing them to create natural pigments [34]. In a similar vein, red pitaya has been utilized as a dye source for making ink and film [35], whereas wood waste has demonstrated its potential to produce durable and effective pigments suitable for paints [36].

From this perspective of environmental-friendly pigments for protective coatings, the potential combination of bio-based pigments with sepiolite powders seems highly intriguing and promising. Clay-based fillers have garnered significant interest because of their remarkable barrier properties, thermal stability, mechanical strength, and ability to resist corrosion [37]. Sepiolite is a type of 2:1 silicate clay mainly made of aggregates of nanorods and nanofiber bundles [38], with an ideal molecular formula of Si_12_Mg_8_O_30_(OH)_4_(OH_2_)_4_-8H_2_O. This material demonstrates a potential for its use in polymers, forming a robust connection with the polymer matrix owing to its exceptional physical attributes and solid thermal stability [39]. As a result, sepiolite has frequently served as a filler to enhance the corrosion resistance of epoxy [40,41,42,43] and polyurethane [44] polymer matrices, or to bolster the thermal insulation within coatings [45,46].

Nevertheless, sepiolite has proven to be highly effective as a protective additive for particular coloring agents. This quality renders it a compelling material for crafting the bio-based pigments utilized in paints. Indeed, drawing inspiration from the Maya Blue pigment, discovered in ancient Maya ruins [47,48], which notably lacks mineral-based color yet exhibits remarkable stability even in highly humid conditions [49], certain researchers explored the adsorption potential of clay minerals. They transformed soluble dyes into insoluble pigments using a liquid-phase method, resulting in hybrid pigments with exceptional stability [50]. In creating a vibrant blue cobalt hybrid pigment, the incorporation of sepiolite not only significantly decreased the cost of the cobalt blue pigment but also enhanced its stability [51]. Subsequent to these findings, considerable research efforts have oriented on investigating the influence of cationic dye species on the structure and functionality of sepiolite hybrid pigments [52,53]. Particularly, there has been a focus on the chemical bonding of indigo with the phyllosilicate material [54,55]. Moreover, multiple recent studies have centered on environmentally friendly methods, such as ball milling [56,57,58] and grinding [59], for the eco-friendly production of the sepiolite-based pigment, omitting the need for chemical reagents.

Despite these premises, there is a lack of comprehensive research in the literature concerning the application of these specific pigments in industrial paint. Hence, this research aims to examine the effectiveness and long-term visual appeal of a sepiolite-based pigment containing *Indigofera tinctoria* extract when incorporated into a water-based primer. *Indigofera tinctoria* L. is a tropical semi-shrub belonging to the *Fabaceae* family [60]. Its leaves serve as the primary source for producing indigo, a highly colorfast blue dye historically referred to as the “king of dyes” due to its exceptional quality [61]. Indigo stands as one of the earliest dyes in human history, utilized across numerous ancient civilizations worldwide [62]. Hence, this extract’s remarkable coloring properties make it well suited for integration into sepiolite fibers, aiming to create a highly effective and durable pigment.

Beyond just aesthetics, this study endeavors to assess how the bio-based pigment influences both the morphology of the coating and its protective properties. Optical microscope and scanning electron microscope (SEM) observations and measurements of color, gloss, and surface roughness were employed to investigate how the bio-based pigment affected both the appearance and structure of the coating. To gauge the impact of varying pigment quantities on the durability of the coating, a range of tests were conducted. These included exposure trials in salt spray and UV-B chambers, assessments of resistance to liquids, contact angle measurements, and electrochemical impedance spectroscopy (EIS) measurements.

## 2. Materials and Methods

### 2.1. Materials

The bio-based pigment was supplied by Pigm’Azur (Nice, France). The pigment manifests as a blue powder comprising 91–92 wt.% sepiolite and 8 wt.% indigo. The indigo molecules are encapsulated in the needle structure of the sepiolite through a simple grinding process. The product particle size averages at 11 µm (D50), with a density of 0.3 g/cm^3^ and a pH ranging between 7 and 8. The waterborne urethane alkyd resin-based white primer paint, Antiruggine RE H_2_O, was supplied by Industrie Bruno Stoppani (Capriano del Colle, BS, Italy). The paint possesses a pH ranging from 8 to 8.5, a specific weight equal to 1.4–1.5 kg/L and a dry residue (% by weight) of 55 ± 3. The carbon steel substrate (Q-panel type R (0.15 wt.% C—Fe bal.)—152 mm × 76 mm × 2 mm dimensions) was purchased by Q-lab (Westlake, OH, USA).

### 2.2. Samples Production

Prior to painting, the steel panels were cleaned using acetone to eliminate any grease and surface dirt, enhancing the bonding between the coating and the surface. Hence, the paint was sprayed on, resulting in a layer approximately 60 µm thick. This initial layer underwent cross-linking at room temperature for 24 h before a second application. As a result, the final coatings have a combined thickness of roughly 120–130 µm.

To assess the pigment’s effectiveness, the paint was prepared by incorporating varying amounts of sepiolite-based powder into two separate solutions. The two specific quantities, 0.5 wt.% and 5.0 wt.%, were deliberately selected for specific reasons. The lower limit of 0.5 wt.% was chosen as it represents the minimum amount required to observe a noticeable color alteration in the coating. On the other hand, the upper limit of 5.0 wt.% was selected because it typically represents the maximum amount commonly utilized in the industrial sector for such purposes. These ranges were chosen to encompass a visible color change while staying within the boundaries commonly employed in industrial practices. Both solutions underwent stirring for 30 min using an ultrasound probe to ensure an even dispersion of the pigment. The performance of these two sets of pigmented coatings was then contrasted with that of a white primer, based solely on a waterborne urethane alkyd resin, devoid of any bio-pigment. Table 1 summarizes the three sample series along with their respective nomenclature, while Figure 1 represents their overall appearance. 

### 2.3. Characterization

The analysis of the pigment’s morphology and the coatings’ cross-sections was conducted using the low vacuum scanning electron microscope (SEM) JEOL IT 300 (JEOL, Akishima, Tokyo, Japan), alongside the optical stereomicroscope Nikon SMZ25 (Nikon Instruments Europe, Amstelveen, The Netherlands). The primary goal was to analyze the distribution of the pigment and ascertain the compatibility between the polymeric matrix and the bio-based powder. Additionally, the influence of varying amounts of sepiolite-based powder on the coatings’ visual appearance was investigated using a Konica Minolta CM2600d spectrophotometer (Konica Minolta, Chiyoda, Tokyo, Japan), employing a D65/10° illuminant/observer configuration in SCI mode. A glossmeter Erichsen 503 (ERICHSEN GmbH & Co. KG, Hemer, Germany), following the ASTM D523-14 (2018) standard [63], was utilized to further examine the coatings’ aesthetic properties. Furthermore, the surface roughness of the coatings was evaluated utilizing the MarSurf PS1 mobile surface roughness measurement instrument (Carl Mahr Holding GmbH, Gottingen, Germany). Furthermore, the coatings’ adhesion was evaluated using a cross-cut test in accordance with the ASTM D3359-17 standard [64]. This assessment aimed to discern whether the incorporation of the pigment had induced any changes in the coatings’ adherence to the steel substrate.

The protective performance of the coatings was investigated through accelerated degradation tests. The samples underwent exposure in a salt spray chamber (Ascott Analytical Equipment Limited, Tamworth, UK) for 500 h, adhering to the ASTM B117-11 standard [65] using a 5 wt.% sodium chloride solution. This test aimed to gauge how the presence of the bio-based pigment influenced the corrosion protection capabilities of the composite layers in a harsh, aggressive environment. Additionally, the adhesion of the coatings was assessed by making a mechanical cut on the sample surface, allowing for the analysis of potential detachment and water uptake phenomena. This evaluation helped determine the coatings’ resilience and adhesion under stress, providing insights into their overall durability and performance.

Likewise, the samples underwent exposure to UV-B radiations (313 nm—60 °C) utilizing a UV173 Box Co.Fo.Me.Gra (Co.Fo.Me.Gra, Milan, Italy), following the ASTM D4587-11 standard [66]. This test spanned a duration of 300 h. To track the coatings’ degradation, colorimetric assessments and gloss analyses were conducted in intervals, specifically, after 24, 48, and every 100 h of UV-B exposure. Moreover, the FTIR spectra of the pigment were acquired in attenuated total reflection (ATR) mode using a Varian 4100 FTIR Excalibur spectrometer (Varian Inc., Santa Clara, CA, USA), before and after the exposure test. This method helped evaluate any chemical modifications occurring in the pigment due to the UV exposure. Furthermore, to better evaluate the aesthetic durability of the pigment for outdoor applications, the coatings were exposed in an oven at temperatures of 100 °C for 24 h, measuring any chromatic changes by means of colorimetric analyses.

The influence of the green pigment on the protective properties of the coating was investigated through cold liquid-resistance tests, following the UNI EN 12720 standard [67]. In this experiment, filter paper was immersed in separate solutions containing 15% sodium chloride, pure acetone, olive oil, and coffee. Moreover, the color resistance of the pigment to acidic and basic conditions was assessed using solutions with pH levels of 1 and 14, respectively, achieved by employing concentrated HCl and NaOH. These soaked filter papers were then placed onto the surface of the coating and covered with a glass lid. After 24 h, the glass cover and filter paper were removed, and any remaining liquid on the coating surface was eliminated. The resulting imprints and any changes in color and gloss were examined using color and gloss analysis. This assessment allowed for the evaluation of the coatings’ resistance to various chemical substances. Moreover, to evaluate the influence of the sepiolite-based pigment on the surface wettability of the coatings, contact angle measurements were performed following the ASTM D7334-08 standard [68]. A Nikon 60 mm lens with an aperture of f/2.8 (Nikon Instruments Europe, Amstelveen, the Netherlands), was used for capturing macro pictures. The contact angle measurements were conducted using the NIS-Elements Microscope Imaging software on a Windows Version platform. Demineralized water droplets (2 µL), generated via a syringe and dispersed from a distance of approximately 2 cm, were observed. Then, 60 s after deposition on the coating, the drop was photographed, and the wetting angle was calculated using the imaging software. To ensure statistical reliability, each sample underwent 10 measurements. This comprehensive approach facilitated a thorough analysis of the coatings’ surface wettability properties, allowing for a more accurate understanding of the impact of the pigment on this aspect.

## 3. Results and Discussion

### 3.1. Pigment Appearance and Coatings Characterization

Figure 2 showcases the bio-based pigment, providing images captured both through the optical microscope and at a more detailed level under the SEM. The optical microscope images offer a macroscopic perspective of the pigment, emphasizing its dark blue hue. Conversely, the SEM images offer a closer inspection, highlighting the distinctive morphological structure of the sepiolite-based pigment. Under SEM observation, the pigment presents as a powder with an average grain size slightly exceeding 10 µm. However, these grains consist of agglomerates comprising extremely short and fine sepiolite fibers, recalling the typical morphology of sepiolite-based pigments [56,58,69]. Within these fiber agglomerations lies the extract of *Indigofera tinctoria* L., contributing to the pigment’s characteristic blue color. Absolutely, the combination of its compact size and vibrant color renders this powder a compelling option as a pigment for organic paints and coatings, particularly those requiring thin applications. EDXS investigations on the pigment revealed the typical composition of sepiolite, with traces of magnesium, aluminum, silicon, potassium, calcium, and iron. The EDXS spectra can be found in Appendix A.

Thus, the addition of the pigment to the white primer resulted in the creation of the series of samples depicted in Figure 1, showcasing the chromatic alteration brought about by the bio-based additive. The initial white hue of the primer transitions towards blue tones, exhibiting a brighter and more pronounced effect depending on the quantity of pigment incorporated. The graph in Figure 3 effectively emphasizes the shift in color, gloss, and roughness induced by the pigment concerning the reference white sample B0.0. It illustrates how the addition of the pigment alters these properties compared to the base white sample, providing a clear visual representation of the changes brought about by the pigment.

The formula used to calculate the color change (ΔE) in the composite coatings is typically derived from the CIELAB color space. It involves the following equation [70]:ΔE = [(ΔL*)^2^ + (Δa*)^2^ + (Δb*)^2^]^1/2^,(1)
where ΔL* represents the difference in lightness, Δa* represents the difference in red–green values and Δb* represents the difference in yellow–blue values. This formula quantifies the overall color difference between the two samples, providing a numerical value (ΔE) that indicates the magnitude of the change in color appearance. Certainly, the color changes induced by the blue pigment, measuring approximately nine and twenty-six points for sample B0.5 and sample B5.0, respectively, can be considered substantial [71]. The initial gloss of sample B0.0, roughly measured at twenty-nine, experiences a decrease to approximately twenty-one and eight for sample B0.5 and sample B5.0, respectively. The pigment not only introduces specific coloration but also heightens the opacity of the primer. This effect is closely tied to a notable escalation in surface roughness (Ra), which transitions from around 0.6 µm for sample B0.0 to 0.9 and 2.3 µm for sample B0.5 and sample B5.0, respectively. The pigment granules, despite their relatively small size, contribute to an elevation in the surface roughness of the composite layer, particularly when added in larger quantities. This increase in roughness correlates with the reduction in gloss, signifying a tradeoff between color intensity and the surface’s smoothness when incorporating higher levels of the pigment.

Absolutely, these investigations undeniably showcase the significant influence of the bio-based pigment on the distinct coloration of the coating, concurrently modifying its texture by notably augmenting roughness while diminishing gloss. Incorporating the sepiolite-based pigment into the paint generates compelling effects, both morphologically and aesthetically, underscoring its capacity to fundamentally transform the coating’s appearance and surface characteristics.

Figure 4 illustrates the observation of samples both in plan and in section under an optical microscope to analyze the impact of the pigment on the coatings’ morphology. Even when present in significant quantities, the pigment demonstrates a homogeneous distribution within the polymeric matrix, facilitating its vibrant coloring effect. Analysis of the sections accentuates the presence of granules, further scrutinized through SEM observations. The section of sample B0.0 displays a homogeneous and compact structure. In contrast, the two composite coatings reveal the presence of sepiolite-based granules firmly integrated into the polymer matrix. Remarkably, these granules do not seem to introduce specific flaws in the coating; instead, they synergistically merge with the polymeric bulk, showcasing excellent compatibility with the urethane alkyd resin. This seamless integration suggests a robust cohesion between the pigment and the polymer matrix, indicating promising properties in terms of structural integrity and compatibility within the coating system.

Even with these considerations, the pigment does indeed impact the coating’s performance, marginally diminishing its adherence to the metal substrate. Figure 5 depicts the outcome of the cross cut test, revealing a decrease in adhesion from grade 5B to 4B observed in sample B0.0 to grade 4B and 3B in sample B0.5 and sample B5.0, respectively [64]. While the pigment granules themselves do not induce substantial defects within the coatings, they do modify the compactness and chemistry of the coating matrix. As a result, this alteration diminishes the robustness of the chemical bond between the urethane alkyd resin and the metal substrate.

Hence, the pigment demonstrates a potent coloring capability that significantly influences the coating’s reflective qualities while inducing substantial alterations in its morphology and texture. Despite its even dispersion within the polymer matrix, the pigment initiates structural modifications at a chemical level, resulting in a decreased adhesion between the coating and the metal substrate. This aspect could potentially impact the coating’s barrier and protective functions. Consequently, various accelerated degradation tests were conducted on the samples to assess how the pigment might affect the durability of the composite coating.

### 3.2. Coatings Durability

#### 3.2.1. Salt Spray Chamber Exposure

The samples underwent monitoring every 24 h for the initial 100 h within the salt spray chamber. Afterward, observations were made at 100 h intervals until the completion of the test. To prompt corrosion reactions at the interface between the substrate and the coating, a 2 mm-wide artificial notch was created on the coating’s surface. Throughout the sample observation period, attention was given to the progression of a blister formation and eventual delamination phenomena of the coating at the notch [72].

Figure 6 illustrates the chronological progression of the degradation morphology close to the notch for every sample set. The intentionally created flaw accelerates the formation of corrosion byproducts, which partially encroach upon the coating. Despite no specific peeling observed, the corrosive damage intensifies gradually. Nevertheless, solely examining the notch does not facilitate the distinction between the behaviors of the three sample sets. All three sample sets demonstrate highly comparable patterns; as anticipated, the emergence of corrosion byproducts facilitates blister development near the notch, irrespective of whether the polymer matrix contains blue pigments or not.

Significant disparities among the samples become evident in regions distant from the artificial defect, where the coating remains undamaged and can showcase its genuine protective capabilities. Even in these areas, blisters form because the primer selected for the study does not possess particularly strong barrier properties. Essentially, the primer’s primary function is to ensure excellent adhesion to the metal substrate. Nonetheless, the examination of pigment behavior was conducted within the primer because of its inherently limited durability. This approach aimed to better emphasize the protective function of the sepiolite-based powder. Figure 7 illustrates a depiction of the blisters that emerged within the three series of samples, situated close to the intact and flaw-free coating. The blisters frequently are accompanied by leaked corrosion products, stemming from the compromised integrity of the coating, no longer effectively shielding the surface. Yet, this occurrence seems less prominent in sample B5.0, where the development of blisters appears to be impacted by the bio-based pigment.

For a thorough examination of the pigment’s behavior, the progression of these blisters was meticulously tracked by subjecting five samples per series to the accelerated degradation test. These analyses encompassed not only the count of blisters that formed over time but also tracked the changes in their average size and the overall area affected by the defect. This area is quantified as the percentage of the sample surface covered by blisters. The findings from these analyses are presented in the three graphs depicted in Figure 8.

To underscore the advantageous impact of the pigment, it is essential to analyze all three graphs concurrently. Figure 8a distinctly emphasizes how the existence of elevated pigment concentrations significantly diminishes the blister count. Specifically, in comparison to the pure urethane alkyd matrix coating, sample B5.0 displays an average of 50% fewer blisters. Simultaneously, Figure 8b indicates that a higher pigment amount generally correspond to larger average blister sizes. However, this trend is intricately linked to the observations in Figure 8a, where a smaller blister count is noted. Specifically, examining the blister size evolution in sample B0.0 and sample B0.5 reveals a distinctive pattern: the blisters undergo rapid initial growth, followed by an average reduction in size between 50 and 200 h of exposure, succeeded by subsequent growth. This decline observed in the second phase actually signifies the emergence of new, smaller blisters. This is supported by the significant rise in slope concerning the increase in blister count, observed in Figure 8a. In sample B5.0, the pattern of blister size fluctuation—growth, decrease, followed by stabilization—is notably absent. Instead, blister sizes tend to increase and reach a point of near stabilization after about 300 h. Analyzing Figure 8a,b together elucidates the behavior of sample B5.0; it demonstrates limited blister development, with subsequent growth over time. Unlike samples B0.0 and B0.5, the high-pigment content coating displays fewer instances of blister development, suggesting either a reduced defectiveness or, at the very least, an effective barrier effect of the pigment in minimizing moisture penetration from the salt spray chamber. These analyses are corroborated by the findings depicted in Figure 8c, affirming that the overall defectiveness in sample B5.0 is notably diminished. Approximately half of the surface area in the samples with high pigment content remains unaffected by the formation of blisters, indicating a substantial reduction in defects compared to the other samples.

In conclusion, it can be confidently stated that the pigment provides a tangible protective benefit, notably seen in its ability to diminish the emergence of defects such as blisters. This effect is prominently noticeable, especially in a non-protective primer akin to the one employed in this study, which exhibits heightened susceptibility to the test solution’s permeation. Nonetheless, this favorable outcome suggests a promising potential for the sepiolite-based powder. It could be applied in multi-layer systems, serving not only as a pigment but also as a protective filler within a top coat, demonstrating its dual functionality and indicating a prospect for further application in enhancing protective coatings.

#### 3.2.2. Outdoor Resistance

Even though alkyd urethane resins are acknowledged for their robust UV resistance [73], the samples underwent an exceptionally intense UV-B radiation exposure to assess the pigment’s long-term visual endurance. Indeed, not just the three series of samples, but the individual pigment was also subjected to the accelerated degradation test. This was aimed at gaining more specific insights into its performance within the polymer matrix and understanding its inherent durability characteristics.

Figure 9 illustrates the visual comparison of the three series of samples and the pigment, observed through an optical microscope before and after the accelerated degradation test. The image emphasizes a shift in the appearance of the three samples, which becomes progressively less noticeable as the pigment quantity increases. Surprisingly, the pigment’s appearance appears relatively unaffected by exposure to UV-B radiation. The polymeric matrix of the coating indeed experiences chemical and physical degradation, noticeable by a slight yellowing, observed in sample B0.0. Nevertheless, it appears that the durability of the pigment mitigates and protects against this phenomenon of aesthetic deterioration.

This observation is elucidated by the graphs depicted in Figure 10, showcasing the changes in color and gloss within the three coating series throughout the test. The UV-B radiation exposure is notably intense, leading to significant degradation of the samples within the initial 24 h of testing. However, afterward, the color and gloss values tend to stabilize. As a result, the test was halted at 300 h of exposure, by which time the samples displayed consistent aesthetic attributes. Figure 10a, which illustrates the color change ΔE during the test, quantitatively substantiates the earlier qualitative observations outlined in Figure 9. As the pigment concentration increases, a noticeable reduction in the color change in the samples are evident: from a final ΔE value of approximately 6.0 points for sample B0.0, it decreases to roughly 3.0 points and 1.5 points for sample B0.5 and sample B5.0, respectively. In this instance, it is conclusive that the color alteration induced by UV-B radiation in sample B5.0 is nearly negligible. The noticeable shift in color in sample B0.0 primarily comes from a rise in the coordinate b*, moving towards the positive values associated with yellow tones. This yellowing is a common occurrence in the photochemical breakdown of polymer structures. Conversely, the b* value stays nearly unchanged in sample B5.0. This affirms the exceptional durability of the bio-based pigment, showcasing its ability to potentially conceal any photochemical degradation occurring in the polymer matrix of the coating. Indeed, the deterioration of the polymer matrix is corroborated by the considerable decrease in gloss observed in sample B0.0 (Figure 10b). However, this gloss reduction is notably less pronounced in the B5.0 sample. This is partially attributed to the pigment’s intrinsic characteristic of introducing a matte effect, linked to the heightened roughness (as depicted in Figure 3). This immediate increase in roughness contributes to lowered reflectance values in the coating.

Ultimately, the incorporation of 5 wt.% of pigment leads to a reduction of about 75% in color change and approximately 71% in gloss change, compared to the coating solely comprised alkyd urethane matrix. These significant reductions highlight not only the pigment’s remarkable color stability but also its contribution to enhancing the overall aesthetic uniformity and consistency of the composite coating.

Yet, to solidify the intriguing durability of the pigment against UV-B radiation-induced degradation, the sepiolite-based powder underwent FTIR analysis both before and after direct exposure in the UV chamber for 300 h. Figure 11 displays the outcomes of this examination, presenting the two spectra of the pigment—one prior to and the other following the accelerated degradation test. The spectrum acquired before the exposure in the UV chamber reveals distinctive peaks representing the composition of the material constituting the bio-based pigment. The spectral bands within the 4000–3000 cm^−1^ range indicate vibrations related to the stretching of the Mg–OH group [74], coordinated water, and zeolitic water present in the compound. Furthermore, the peak observed at 1619 cm^−1^ specifically signifies the vibrational pattern characteristic of zeolitic water [75]. Other notable peaks include those at 1199 cm^−1^ and 972 cm^−1^, attributed to Si–O bonds, and the peak at 1011 cm^−1^, indicative of Si–O–Si plane vibrations [75,76]. Lastly, the signal detected at 690 cm^−1^ corresponds to the vibrations associated with the bending vibration of Mg–OH [74]. The indigo extract signal, while existing in smaller amounts within the powder, is significantly masked by the dominant sepiolite signature. Specifically, the characteristic peaks of indigo, such as those linked to C=O vibrations near 1600 cm^−1^, 1060 cm^−1^, and 690 cm^−1^ [77], as well as the stretching vibration of the C–N group combined with the rocking of C–H and N–H groups around 1170 cm^−1^ [78], overlap and coincide with the signals attributed to sepiolite that were previously described.

The second spectrum on the graph, representing the pigment’s condition after the accelerated degradation tests, notably overlays perfectly with the first one. This emphasizes the consistency and chemical stability of the sepiolite-based powder. Therefore, the analysis affirms that exposure to UV-B radiation has no discernible impact on the pigment’s appearance, indicating minimal degradation. 

Indeed, sepiolite’s strong UV resistance [51,79,80] was harnessed during the pigment’s design to enhance the visual endurance of the Indigofera tinctoria L. extract, which is well known for its susceptibility to photofading of color induced by UV radiation [81,82], encompassing the cleavage of C=C double bonds [83].

As a result, the UV chamber exposure test underscored the outstanding performance of the bio-based pigment, showcasing its ability to sustain prolonged color stability without succumbing to photodecay caused by UV-B radiation. Moreover, the pigment demonstrated its capacity to minimize the aesthetic degradation of the coating’s polymeric matrix, preserving the overall appearance nearly unchanged throughout the test.

However, a crucial consideration for pigments employed in outdoor applications is the exposure to high temperatures that surfaces may experience during the hottest seasons. Thus, to assess the chromatic stability of the pigment under elevated temperatures, the color and gloss of the samples were evaluated after a 24 h exposure at a temperature of 100 °C. Figure 12 illustrates the variation in these two parameters following the test.

The two graphs highlight a good chromatic consistency provided by the pigment even at high temperatures, as both the gloss and the color of the samples show slight changes following the accelerated degradation test. Specifically, sample B0.0 demonstrates a color change ΔE of approximately 5 points. Notably, the color stability of the samples improves with higher pigment concentrations. Consequently, while the coating lacking the bio-based pigment experiences a slight color shift when exposed to high temperatures, the sepiolite-based additive successfully preserves consistent color. This phenomenon is ascribed to the well-known thermal durability of sepiolite [84,85]. Sepiolite effectively shields the indigo extract, with which it is infused, from potential thermal degradation phenomena. Similarly, the delta gloss values remain relatively low, affirming the samples’ good aesthetic uniformity.

In essence, the pigment proves its suitability for outdoor applications, which often pose the greatest challenges for natural and bio-based pigments [30,32,86]. Its exceptional resistance enables the sepiolite-based powder to effectively shield the coloring extract within, maintaining its vibrant blue hue intact.

#### 3.2.3. Liquids Resistance

The two different amounts of the sepiolite-based pigment were analyzed to see how they impacted the barrier properties of the polymeric matrix across a range of tests. The outcomes from the cold liquid resistance test are presented in Figure 13. Figure 13a displays the changes in color, referred to as ΔE, following interactions between the coatings and the four test solutions. Meanwhile, Figure 13b illustrates how the surface gloss of the samples changed based on the chosen test solution. As expected, the NaCl solution typically does not cause significant color changes in the coating, resulting in ΔE values that fall within the category 0 range, according to standard [87]. On the contrary, the gloss appears to be influenced by exposure to the saline solution. Yet, the decrease in gloss becomes less conspicuous with higher levels of pigment. The acetone yields comparable results in both color alteration and gloss: the former being minimal, while the latter diminishes in relation to the presence of the bio-based pigment. Regarding color alteration, the oil showcases an almost imperceptible impact while causing a roughly 10-point rise in gloss across all three groups of samples. This pattern is characteristic of a highly reflective substance like olive oil [88]. Lastly, coffee has the least effect on the gloss of the coatings but induces the most notable color alteration. Based on the reference standard [87], samples B0.0, B0.5, and B5.0 demonstrate ΔE values corresponding to categories 4, 2-3, and 1, respectively. The positive impact of the pigment on the long-term appearance of the coating is undeniable, evident in the superior color and reflectance stability of sample B5.0. This effect is likely aided by the initial darker shades that partially obscure the absorption of dark solutions within the coating. Indeed, this finding aligns with the observations from the salt spray chamber test, affirming a positive barrier effect of the pigment. This effect complicates the percolation of the test solution into the coating of the B5.0 sample, as depicted in Figure 8. Additionally, the higher surface roughness and consequently lower initial gloss values make sample B5.0 less susceptible to alteration when in contact with test solutions. Likewise, sepiolite has a tendency to minimize color changes caused by highly acidic or alkaline solutions. For instance, sample B5.0 demonstrates minimal alterations in both color and gloss when exposed to pH 1 and pH 14 solutions. Despite indigo’s vulnerability to varying acidity levels [89], sepiolite maintains the pigment’s color consistency, shielding it from the typical degradation processes.

Surface roughness is a pivotal factor in enhancing the long-lasting color stability of the coating, closely linked with a hydrophobic effect. Figure 14 displays the outcomes of the contact angle measurements conducted on the three sample types, including the average contact angle value derived from measurements of 10 distinct drops per sample series. The analysis underscores a distinct pattern: a rise in the contact angle corresponding to the quantity of pigment integrated into the paint. This trend is not directly linked to the inherent hydrophobic nature of the pigment but rather to an escalation in surface roughness facilitated by the sepiolite-based powder, as demonstrated earlier in Figure 3. Indeed, various studies highlight how surface roughness can influence the hydrophobic-hydrophilic traits of a surface [30,90,91]. Nevertheless, while the addition of 5 wt.% of pigment leads to a 25% increase in the contact angle compared to the reference sample B0.0, it is inaccurate to label these coatings as purely hydrophobic, since a 73° contact angle is not indicative of a distinctly hydrophobic surface. Absolutely, the pigment, by changing the surface texture of the coating, indeed adjusts its hydrophobic–hydrophilic properties, thus affecting its resilience when exposed to specific aggressive solutions, as emphasized by the graphs in Figure 13.

In summary, it can be inferred that the sepiolite-based pigment does not encourage specific percolation or solution absorption within the acrylic matrix of the coating. Instead, it appears to bolster the coating’s ability to act as a barrier, diminishing color fading and gloss reduction while enhancing its hydrophobic traits. Consequently, these discoveries shed light on potential novel applications for this environmentally friendly material as an innovative pigment in organic coatings, capable of imparting unique aesthetic qualities and heightened resistance to liquids.

## 4. Conclusions

This study underscores the potential of an innovative bio-based pigment derived from sepiolite, infused with *Indigofera tinctoria* L. extract, offering applicability in exterior paints. This pigment, displaying a vibrant blue hue when incorporated into a white primer, notably transforms its visual appearance by introducing colors that vary according to the powder concentration. Additionally, it diminishes the gloss and enhances the surface roughness of the composite layer.

The pigment, uniformly dispersed within the polymer matrix, induces a minor shift in its structural chemistry, leading to a slight reduction in its coating adhesion. However, this phenomenon does not compromise the protective capabilities of the composite layer; rather, it enhances them. Several accelerated degradation tests have consistently demonstrated the pigment’s exceptional role in augmenting the durability of the white primer.

Exposure within the salt spray chamber has underscored the pigment’s significant barrier properties, effectively restricting solution absorption within the coating and subsequently minimizing the occurrence of visible defects. Simultaneously, the UV-B radiation exposure test has revealed the pigment’s intriguing capacity to maintain prolonged color stability, demonstrating resilience against the photodegradation phenomena over an extended period. Likewise, the thermal resilience of sepiolite serves to safeguard the pigment against thermal degradation, thereby preserving the aesthetic attributes of the coating even after exposure to elevated temperatures. Finally, the liquid resistance tests and contact tangle measurements have underscored the pigment’s pivotal role in fortifying the coating’s barrier function. This includes mitigating color fading, reducing gloss loss, and enhancing the coating’s hydrophobic characteristics.

Ultimately, the pigment has consistently demonstrated its ability to enhance the durability of the coating by mitigating the absorption of aggressive solutions. Specifically, the sepiolite-based powder complicates the percolation path of aggressive solutions within the polymeric matrix of the coating, which would typically absorb liquids due to its intrinsic porosity. Moreover, the pigment alters the surface texture of the coating, thereby slightly augmenting its hydrophobic properties.

These findings collectively reveal promising prospects for this environmentally friendly material as an innovative pigment in organic coatings. Its capability to bestow distinctive aesthetic attributes while augmenting the coatings durability opens doors to novel applications in various industries.

## Figures and Tables

**Figure 1 materials-17-00941-f001:**
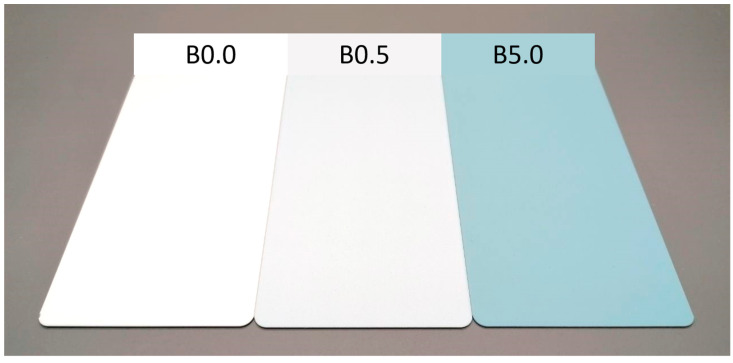
Samples appearance (panel widths equal to 76 mm).

**Figure 2 materials-17-00941-f002:**
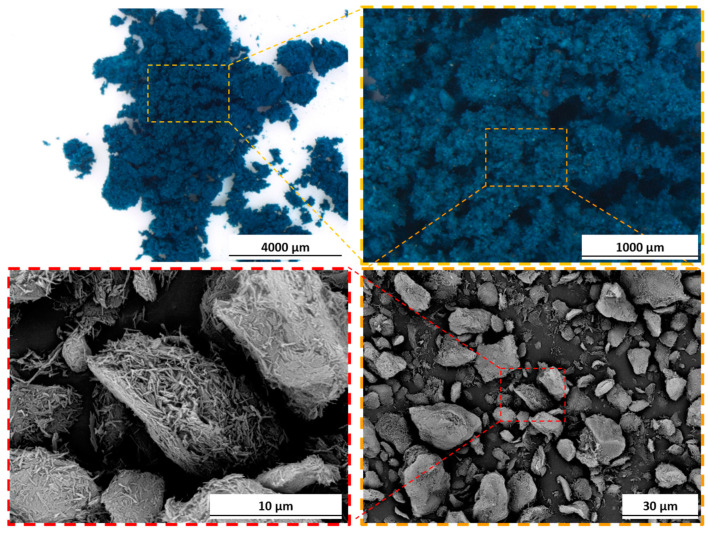
Optical microscope and SEM images of bio-based pigment.

**Figure 3 materials-17-00941-f003:**
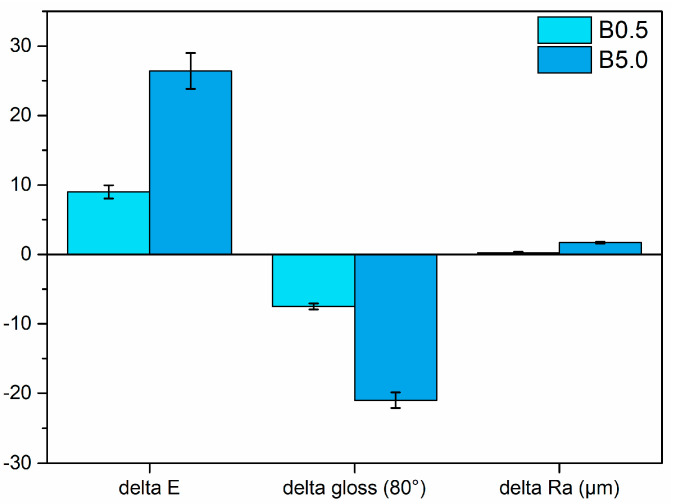
Change in color, gloss, and roughness with respect to the reference sample B0.0.

**Figure 4 materials-17-00941-f004:**
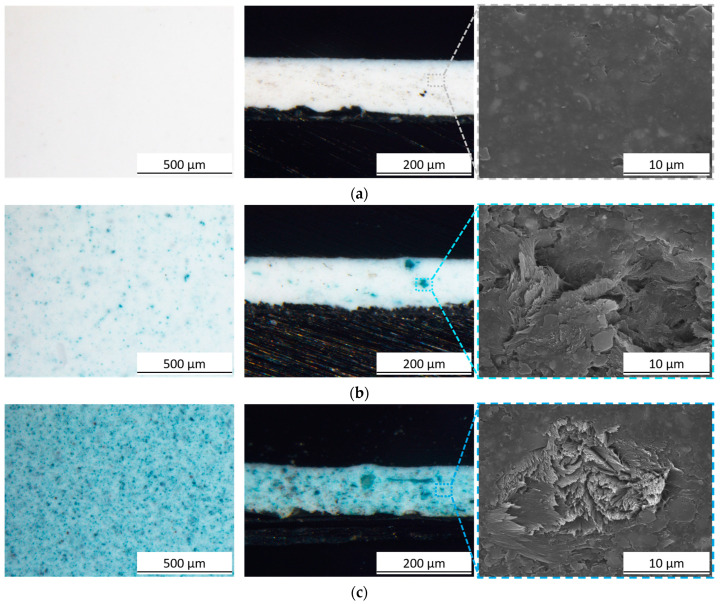
Optical micrographs of the top view (on the left) and cross-section (on the right) of (**a**) sample B0.0, (**b**) sample B0.5 and (**c**) sample B5.0. The images showcase a SEM-generated focus, emphasizing the internal composition of the coatings.

**Figure 5 materials-17-00941-f005:**

Cross cut test results of (**a**) sample B0.0, (**b**) sample B0.5 and (**c**) sample B5.0, observed with optical microscope.

**Figure 6 materials-17-00941-f006:**
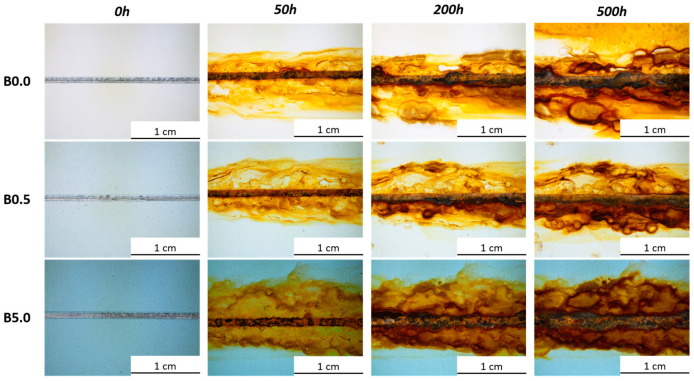
The degradation of the coatings near the artificial defect, as a function of the exposure time in the salt spray chamber.

**Figure 7 materials-17-00941-f007:**
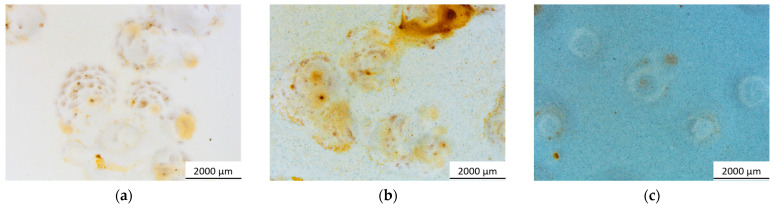
Example of blisters observed on the three series of samples, where (**a**) represents sample B0.0, (**b**) sample B0.5, and (**c**) sample B5.0.

**Figure 8 materials-17-00941-f008:**
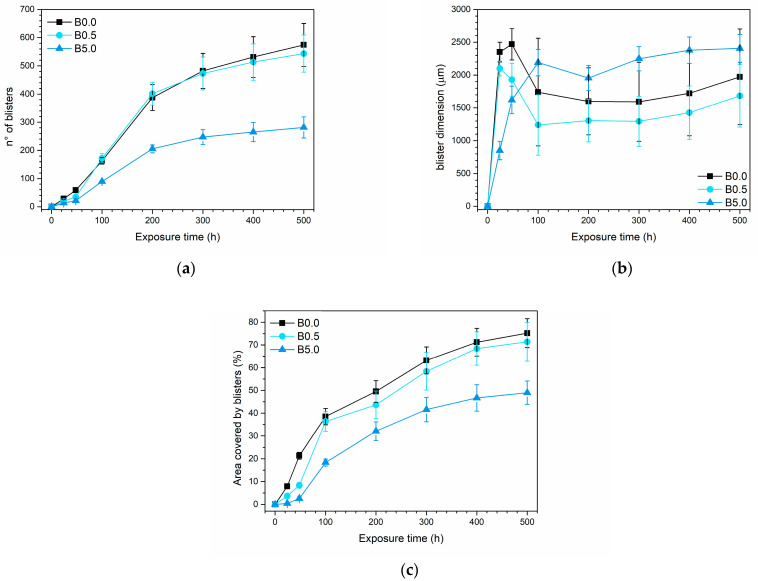
Changes observed in (**a**) blister quantity, (**b**) average blister size, and (**c**) the area occupied by blisters throughout sample exposure within the salt spray chamber.

**Figure 9 materials-17-00941-f009:**
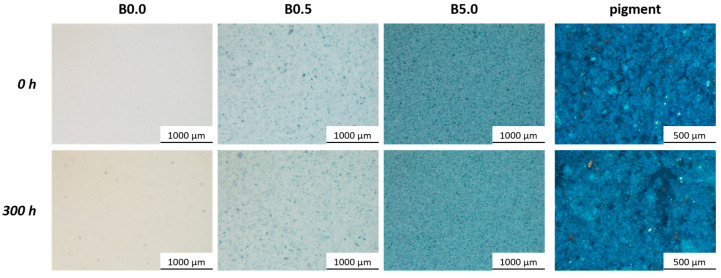
The transformation in the appearance of the three sample series and the pigment subsequent to the UV-B radiation exposure test.

**Figure 10 materials-17-00941-f010:**
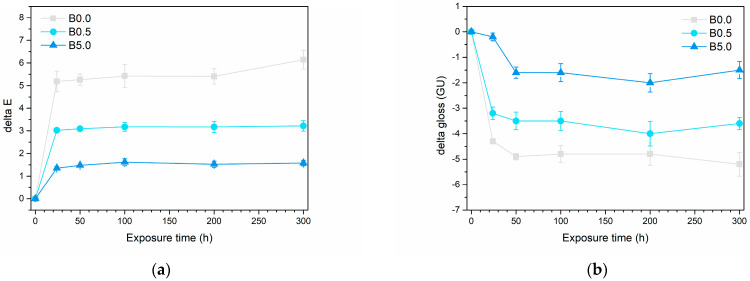
Evolution of (**a**) color and (**b**) gloss during the UV-B exposure test.

**Figure 11 materials-17-00941-f011:**
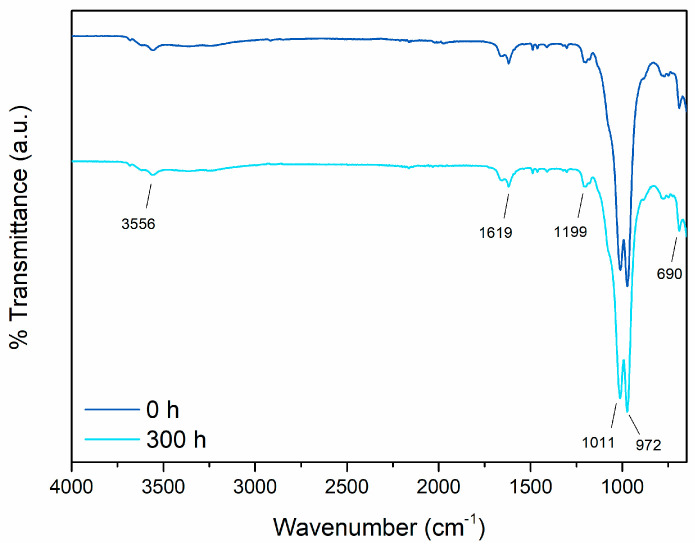
FTIR spectra of the bio-based pigment before and after exposure to UV-B radiation.

**Figure 12 materials-17-00941-f012:**
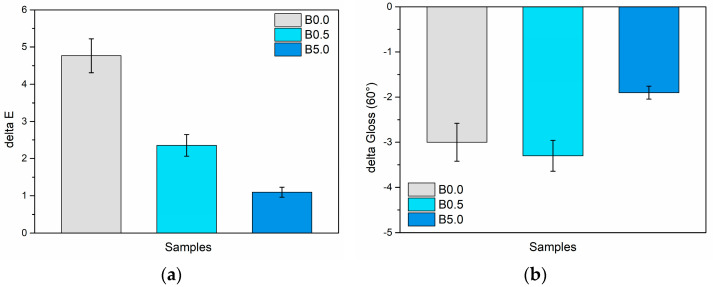
Change in (**a**) color and (**b**) gloss due to the thermal exposure test.

**Figure 13 materials-17-00941-f013:**
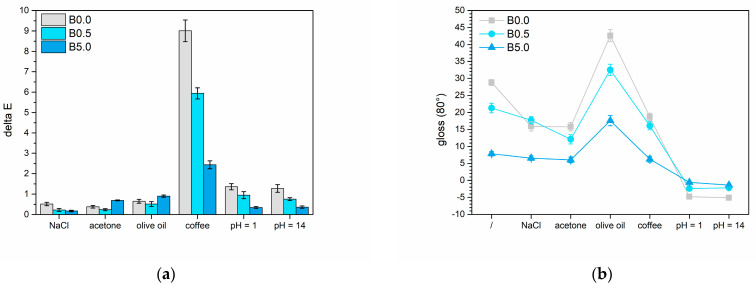
Variation in (**a**) color and (**b**) gloss of the samples after the liquid resistance test.

**Figure 14 materials-17-00941-f014:**
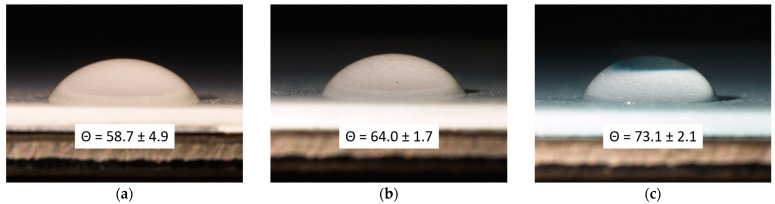
Optical micrograph of the contact angle measurements of (**a**) sample B0.0, (**b**) sample B0.5 and (**c**) sample B5.0, with respective average contact angle values.

**Table 1 materials-17-00941-t001:** Samples nomenclature, with associated amount of bio-based pigment.

Sample Nomenclature	Bio-Based Pigment Concentration (wt.%)
B0.0	0.0
B0.5	0.5
B5.0	5.0

## Data Availability

The data presented in this study are available on request from the corresponding author. The data are not publicly available due to the absence of an institutional repository.

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
