# Peer review of "Assessing the Impact of Sepiolite-Based Bio-Pigment Infused with Indigo Extract on Appearance and Durability of Water-Based White Primer"

_materials, 2024, doi:10.3390/ma17040941_

Round 1

Reviewer 1 Report

Comments and Suggestions for Authors

1. For Figure 8, the two light colored lines (light blue and light gray) in the figure are not clear, so it is recommended to replace them with more eye-catching colors.

2. For Line436, cm cm-1, a cm should be deleted.

3. Essays must be written in SI units. For example, hour should be changed to h.

4. The addition of sepiolite based pigment powder significantly improves the surface roughness of the coating and reduces the bonding of the coating. When the amount of addition reaches what level (more than 5%), the coating bond is so poor that it is not usable?

5. What challenges does the addition of sepiolite based pigment powder pose to the coating preparation process? Does this challenge affect its practical application?

6. How much indigo is added to 5% compound powder? What are the advantages of the coating performance compared with the addition of this amount of indigo pure pigment (without sepiolite)?

7. Line519, what is the experimental evidence for the slight change in structural chemistry claimed by the author? If so, it should be added to the paper.

8. The addition of sepiolite powder is conducive to protecting the coating (improving durability). What is the microscopic mechanism of this protection?

Comments on the Quality of English Language

1. For Line436, cm cm-1, a cm should be deleted.

2. Essays must be written in SI units. For example, hour should be changed to h.

Author Response

  1. For Figure 8, the two light colored lines (light blue and light gray) in the figure are not clear, so it is recommended to replace them with more eye-catching colors.

Authors: the pictures have been changed, with darker colors for sample B0.0 and sample B0.5.

  1. For Line436, cm cm-1, a cm should be deleted.

Authors: the authors thank the reviewer for the comment. The typo has been corrected.

  1. Essays must be written in SI units. For example, hour should be changed to h.

Authors: the authors followed the advice of the reviewer, changing hour in h.

  1. The addition of sepiolite based pigment powder significantly improves the surface roughness of the coating and reduces the bonding of the coating. When the amount of addition reaches what level (more than 5%), the coating bond is so poor that it is not usable?

Authors: The reviewer's inquiry poses an intriguing point, yet the authors lack precise knowledge on the weight percentage (wt.%) of pigment that leads to a significant decrease in adhesion between the coating and substrate. The authors explored two distinct pigment concentrations, with 5 wt.% being deemed the upper threshold commonly employed in industrial paint formulations. Given that even at these higher concentrations, the adhesion of the B5.0 coating remains adequate, the authors opted not to delve into the impact of additional pigment addition. This decision was driven by its lack of practical utility and, more importantly, its questionable feasibility for potential commercial utilization. In fact, at line 137 the authors already wrote: “On the other hand, the upper limit of 5.0 wt.% was selected because it typically represents the maximum amount commonly utilized in the industrial sector for such purposes. These ranges were chosen to encompass a visible color change while staying within the boundaries commonly employed in industrial practices.”

  1. What challenges does the addition of sepiolite based pigment powder pose to the coating preparation process? Does this challenge affect its practical application?

Authors: As described on line 141, the addition of the pigment required the use of a 30-min treatment with an ultrasound probe to homogeneously distribute the pigment in the paint: “Both solutions underwent stirring for 30 minutes using an ultrasound probe to ensure an even dispersion of the pigment.” This issue doesn't pose a major obstacle, as the formulation of pigmented paints typically involves procedures such as ultrasound treatment or the use of high-performance stirrers. Consequently, incorporating this pigment, even in significant concentrations, remains highly compatible with coloring paints.

  1. How much indigo is added to 5% compound powder? What are the advantages of the coating performance compared with the addition of this amount of indigo pure pigment (without sepiolite)?

Authors: The reviewer's question is certainly very interesting and correct. However, the composite pigment used in this study was kindly provided by a company, which holds the technical specifications and which needs to maintain industrial secrecy regarding the quantities of the two materials (indigo extract and sepiolite) used in the formulation of the pigment. The authors therefore do not know exactly how much indigo the pigment contains, just as they cannot make a real comparison of the performance that the pigment would have in the absence of sepiolite. In the absence of these specific data, the authors avoid studying the behavior of sepiolite alone, or of indigo extract alone, because they would not be comparable with the real qualities contained in the composite pigment.

  1. Line519, what is the experimental evidence for the slight change in structural chemistry claimed by the author? If so, it should be added to the paper.

Authors: the authors speak of a slight change in structural chemistry as they observed a slight decrease in the adhesion of the coating in the presence of high quantities of pigment. This aspect is clearly due to the fact that 5 wt.% of additive leads to structural and chemical changes in the bulk of the coating, such as discontinuities in the polymer chain of the matrix (see Figure 4). In fact, this aspect is also closely linked to a different behavior of the samples exposed, for example, in salt spray chambers and UV chambers. Therefore, the authors have amply demonstrated how the addition of the pigment influences the structure of the coating, with significant consequences regarding its durability performance.

  1. The addition of sepiolite powder is conducive to protecting the coating (improving durability). What is the microscopic mechanism of this protection?

Authors: the pigment has been shown on several occasions to improve the durability of the coating, reducing phenomena of absorption of aggressive solutions. In fact, the sepiolite-based powder makes the percolation path of the aggressive solution within the polymeric matrix of the coating more complex (whose normal porosity instead leads to easy absorption of liquids). Furthermore, the pigment involves a modification of the surface texture of the coating, slightly increasing its hydrophobic characteristics. The authors had written at line 370: “In conclusion, it can be confidently stated that the pigment provides a tangible protective benefit, notably seen in its ability to diminish the emergence of defects such as blisters. This effect is prominently noticeable, especially in a non-protective primer akin to the one employed in this study, which exhibits heightened susceptibility to the test solution's permeation.” Moreover, at 512: “Absolutely, the pigment, by changing the surface texture of the coating, indeed adjusts its hydrophobic-hydrophilic properties, thus affecting its resilience when exposed to specific aggressive solutions..”

The authors added the following sentences at line 546: “Ultimately, the pigment has consistently demonstrated its ability to enhance the durability of the coating by mitigating the absorption of aggressive solutions. Specifically, the sepiolite-based powder complicates the percolation path of aggressive solutions within the polymeric matrix of the coating, which would typically absorb liquids due to its intrinsic porosity. Moreover, the pigment alters the surface texture of the coating, thereby slightly augmenting its hydrophobic properties.”

Reviewer 2 Report

Comments and Suggestions for Authors

I have read the article "Assessing the Impact of Sepiolite-Based Bio-Pigment Infused with Indigo Extract on Appearance and Durability of Water-Based White Primer" with great interest.  It highlights the potential of an innovative bio-based pigment derived from sepiolite infused with Indigofera tinctoria L. extract, showcasing its applicability in exterior paints. However, specific crucial aspects require additional attention. Hence, I recommend that the publication of this work be considered after the authors address these significant details.

1.      Bio-based materials have also been used for durable liquid repellent coating. Authors are suggested to discuss it in the introduction and cite the following article: Reduction of imine-based cross-linkages to achieve sustainable underwater superoleophobicity that performs under challenging conditions. Journal of Materials Chemistry A 8 (30), 15148-15156.

2.      One of the advantages of this work is the absence of volatile organic solvents, with the utilization of waterborne urethane alkyd resin-based white primer paint. Authors are suggested to provide a more concise discussion of its eco-friendly aspects in the introduction part and cite the following important articles related to it: Bio-Based Waterborne Polyurethane Coatings with High Transparency, Antismudge and Anticorrosive Properties. ACS Appl. Mater. Interfaces 2023, 15, 5, 7427–7441.

3.      Does the color of the coating remain intact after prolonged exposure to acidic and basic conditions?

4.      Authors are suggested to provide EDX (Energy-dispersive X-ray spectroscopy) mapping of the coating to gain more clarity on the chemical composition of samples B0.0, B0.5, and B5.0.

5.      More structural analysis of this material is required. Please provide X-ray diffraction (XRD) patterns for this material.

6.      What is the thermal stability of the material? Please provide thermogravimetric (TG) and derivative thermogravimetric (DTG) curves for the material.

7.      What transformation in the appearance of the samples occurs with elevated temperature?

Comments on the Quality of English Language

Minor editing of English language required

Author Response

I have read the article "Assessing the Impact of Sepiolite-Based Bio-Pigment Infused with Indigo Extract on Appearance and Durability of Water-Based White Primer" with great interest.  It highlights the potential of an innovative bio-based pigment derived from sepiolite infused with Indigofera tinctoria L. extract, showcasing its applicability in exterior paints. However, specific crucial aspects require additional attention. Hence, I recommend that the publication of this work be considered after the authors address these significant details.

  1. Bio-based materials have also been used for durable liquid repellent coating. Authors are suggested to discuss it in the introduction and cite the following article: Reduction of imine-based cross-linkages to achieve sustainable underwater superoleophobicity that performs under challenging conditions. Journal of Materials Chemistry A 8 (30), 15148-15156.

Authors: the authors added the following sentences at line 55: ”Similarly, numerous studies have examined the impact of bio-based resources as environmentally friendly additives to enhance the hydrophobic characteristics of organic coatings. This exploration involves the utilization of natural waxes [29,30] and animal proteins [31], among other green alternatives.”

  1. One of the advantages of this work is the absence of volatile organic solvents, with the utilization of waterborne urethane alkyd resin-based white primer paint. Authors are suggested to provide a more concise discussion of its eco-friendly aspects in the introduction part and cite the following important articles related to it: Bio-Based Waterborne Polyurethane Coatings with High Transparency, Antismudge and Anticorrosive Properties. ACS Appl. Mater. Interfaces 2023, 15, 5, 7427–7441.

Authors: the authors do not entirely agree with the reviewer's comment, as the subject of the study is the bio-based pigment, rather than the waterborne matrix of the coating. Of course, it was chosen to use a solvent-free matrix, to study a system that was as 'green' as possible, but the authors believe it would be misleading to further lengthen the introduction by dealing with the topic of eco-friendly matrices for paints.

  1. Does the color of the coating remain intact after prolonged exposure to acidic and basic conditions?

Authors: In Figure 8, the authors have emphasized the outcomes of the liquid resistance test, conducted in accordance with the specifications outlined in the UNI EN 12720 standard - "Assessment of surface resistance to cold liquids," a widely utilized method for assessing the resilience of pigmented coatings when in contact with various liquids. The selection of the four liquids tested was guided by the standard, aiming to encompass the most common and aggressive liquids, particularly concerning the chromatic stability of the pigments. Consequently, while some liquids, such as coffee, exhibit acidic pH levels, the authors did not specifically focus on the acidity or basicity aspects of the liquids under examination. However, the authors decided to follow the reviewer's advice, implementing the liquid resistance test with solutions at pH 1 and pH 14. The authors added the following sentence to line 309: “Moreover, the color resistance of the pigment to acidic and basic conditions was assessed using solutions with pH levels of 1 and 14, respectively, achieved by employing concentrated HCl and NaOH.” The authors changed the images in Figure 13, and added the following sentences at line 612: “Likewise, sepiolite has a tendency to minimize color changes caused by highly acidic or alkaline solutions. For instance, sample B5.0 demonstrates minimal alterations in both color and gloss when exposed to pH 1 and pH 14 solutions. Despite indigo's vulnerability to varying acidity levels [89], sepiolite maintains the pigment's color consistency, shielding it from typical degradation processes.”

  1. Authors are suggested to provide EDX (Energy-dispersive X-ray spectroscopy) mapping of the coating to gain more clarity on the chemical composition of samples B0.0, B0.5, and B5.0.

Authors: the authors hold differing views regarding the necessity of presenting EDXS maps of the coatings. They argue that the distribution of the pigment is already clearly discernible from the images in Figure 4. In these images, the optical microscope analyses vividly depict the well-homogeneous distribution of the blue pigment throughout the coating's interior, while the sectional SEM analyses reveal the morphology of the pigment integrated into the sample's bulk. The addition of mappings would not offer further indications than those already provided by the images in Figure 4, but would represent a sort of repetition that would further lengthen the text. However, the authors have added an EDXS analysis of the pigment in the Appendix section, in addition to the following sentence on line 286: “EDXS investigations on the pigment revealed the typical composition of sepiolite, with traces of magnesium, aluminum, silicon, potassium, calcium, and iron. The EDXS spectra can be found in Appendix A.”

  1. More structural analysis of this material is required. Please provide X-ray diffraction (XRD) patterns for this material.

Authors: the authors hold the opinion that showcasing an XRD analysis of the pigment is unnecessary since it wouldn't yield additional insights into the material beyond observing the typical structure of sepiolite. Moreover, such an analysis would be more meaningful if the authors had both the initial sepiolite and the one infused with the indigo extract, allowing them to assess the impact of the indigo addition on the morphological structure of sepiolite. However, as they only possess the final pigment, conducting a simple EDXS analysis wouldn't provide useful information. The authors hope that the reviewer can agree with them.

  1. What is the thermal stability of the material? Please provide thermogravimetric (TG) and derivative thermogravimetric (DTG) curves for the material.

Authors: Once again, the authors contend that the requested analysis would be misleading for the objectives of the study, which primarily focus on assessing the functionality of this bio-based pigment for paints. They argue that TG and DTG analyses of the material are not pertinent to the aims of the study or to the application of the pigment. The thermal stability of the pigment is not considered crucial, as it is intended for applications where the paint is not subjected to significant thermal fluctuations.

  1. What transformation in the appearance of the samples occurs with elevated temperature?

Authors: As with the previous question, the authors consider it not functional to the study to evaluate the chromatic stability of the pigment at high temperatures, as it is certainly not designed for this purpose. However, the authors tried to follow the reviewer's advice, evaluating the color change of the samples following exposure to an oven at 100°C for 24 hours. The authors added the following sentences at line 253: “Furthermore, to better evaluate the aesthetic durability of the pigment for outdoor applications, the coatings were exposed in an oven at temperatures of 100°C for 24 h, measuring any chromatic changes by means of colorimetric analyses.” Thus, the authors added a new Figure 12, showing the change in color and gloss after the thermal exposure test, together with subsequent analysis of the chromatic durability of the samples at high temperatures.

The authors added the sentences at line 565: “However, a crucial consideration for pigments employed in outdoor applications is the exposure to high temperatures that surfaces may experience during the hottest seasons. Thus, to assess the chromatic stability of the pigment under elevated temperatures, the color and gloss of the samples were evaluated after a 24-h exposure at a temperature of 100°C. Figure 12 illustrates the variation in these two parameters following the test. The two graphs highlight a good chromatic consistency provided by the pigment even at high temperatures, as both the gloss and the color of the samples show slight changes following the accelerated degradation test. Specifically, sample B0.0 demonstrates a color change ΔE of approximately 5 points. Notably, the color stability of the samples improves with higher pigment concentrations. Consequently, while the coating lacking the bio-based pigment experiences a slight color shift when exposed to high temperatures, the sepiolite-based additive successfully preserves consistent color. This phenomenon is ascribed to the well-known thermal durability of sepiolite [84,85]. Sepiolite effectively shields the indigo extract, with which it is infused, from potential thermal degradation phenomena. Similarly, the delta gloss values remain relatively low, affirming the samples' good aesthetic uniformity.”

The authors also added the following sentences in the conclusions: “Likewise, the thermal resilience of sepiolite serves to safeguard the pigment against thermal degradation, thereby preserving the aesthetic attributes of the coating even after exposure to elevated temperatures.”

Round 2

Reviewer 1 Report

Comments and Suggestions for Authors

The quality of the manuscript has greatly improved and is ready for publication.

Comments on the Quality of English Language

The quality of the manuscript has greatly improved and is ready for publication.

Reviewer 2 Report

Comments and Suggestions for Authors

Thank you for all the comments.